

# Evaluation of isotopes and elements in planktonic foraminifera from the Mediterranean Sea as recorders of seawater oxygen isotopes and salinity

Linda K. Dämmer[1], Lennart de Nooijer[1], Erik van Sebille[2], Jan G. Haak[1], Gert-Jan Reichart[1, 3]

[1]Department of Ocean Systems, NIOZ Royal Netherlands Institute for Sea Research, and Utrecht University, Texel, The Netherlands
[2]Department of Physics, Institute for Marine and Atmospheric research Utrecht (IMAU), Utrecht University, Utrecht, The Netherlands
[3]Department of Earth Sciences, Faculty of Geosciences, Utrecht University, Utrecht, The Netherlands

*Correspondence to*: Linda K. Dämmer (Linda.Daemmer@nioz.nl)

**Abstract.** The Mediterranean Sea is characterized by a relatively strong west to east salinity gradient, which makes it an area suitable to test the effect of salinity on foraminiferal shell geochemistry. We collected living specimens of the planktonic foraminifer *Globigerinoides ruber* (white) to analyse the relation between element/Ca ratios, stable oxygen isotopes of their shells and surface seawater salinity, isotopic composition and temperature. The oxygen isotopes of sea surface water correlate with salinity in the Mediterranean also during winter, when sampled for this study. Sea water oxygen and hydrogen isotopes are positively correlated in both the eastern and western Mediterranean Sea, though especially in the eastern part the relationship differs from values reported previously for that area. The slope between salinity and seawater oxygen isotopes is lower than previously published. Still, despite the rather modest slope, seawater and foraminiferal carbonate oxygen isotopes are correlated in our dataset although with large residuals and high residual variability. This scatter can be due to either biological variability in vital effects or environmental variability. Numerical models backtracking particles show ocean current driven mixing of particles of different origin might dampen sensitivity and could result in an offset caused by horizontal transport. Results show that Na/Ca is positively correlated to salinity and independent of temperature. Foraminiferal Mg/Ca increases with temperature, as expected, and in line with earlier calibrations, also in the high salinity environment. By using living foraminifera during winter, the previously established Mg/Ca-temperature calibration is extended to temperatures below 18 °C, which is a fundamental prerequisite of using single foraminifera for reconstructing past seasonality.

## 1 Introduction

Ocean circulation plays an important role in Earth's climate, by redistributing heat and also by impacting global biogeochemical cycles. Seawater temperature and salinity are key parameters for reconstructing ocean circulation, since together they determine seawater density and thereby largescale circulation patterns, including a substantial part of meridional overturning circulation. Reconstruction of past ocean environments largely relies on so-called proxy calibrations in which a





variable which can be measured in the geological record is related to a target environmental parameter. The incorporation of trace metals in foraminiferal shell carbonate, for example, is a popular tool to reconstruct past ocean parameters. More specifically, the incorporation of Mg (often expressed as the calcite's Mg/Ca) increases exponentially with seawater

temperature, as first observed in culture studies (Nürnberg et al., 1996) and later confirmed by field calibrations (Anand et al., 2003).

In addition to temperature, salinity and inorganic carbon chemistry also affect Mg/Ca in some species of foraminifera (Allison et al., 2011; Dueñas-Bohórquez et al., 2011; Geerken et al., 2018; Gray et al., 2018; Hönisch et al., 2013; Kisakürek et al., 2008; Lea et al., 1999). For the best possible accuracy such effects need to be corrected for when using foraminiferal Mg/Ca

for the reconstruction of temperature, which calls for independent proxies for these other environmental parameters.

Currently, salinity is often reconstructed through indirect relationships with other variables, such as the ratio of stable oxygen isotopes of sea water, which are recorded in planktonic foraminifera (Rohling, 2007), although direct approaches have also been suggested recently (Bertlich et al., 2018; Wit et al., 2013). Since seawater oxygen isotope ratio and salinity are both affected by evaporation and precipitation, the two often are linearly related (Bahr et al., 2013; Rohling, 2007), with their

calibration depending on local conditions. If foraminifera precipitate their calcite in equilibrium with respect to sea water oxygen isotopes, their $\delta^{18}O$ should reflect that of the seawater, and hence salinity. However, as seawater temperature affects stable oxygen isotope fractionation during calcification (McCrea, 1950; Urey et al., 1951) independent temperature reconstructions are needed to estimate seawater $\delta^{18}O$ from $\delta^{18}O_{calcite}$ (Rohling, 2007). Independent temperature reconstructions can be based for example on organic proxies such as $U^{K'}_{37}$ (Prahl and Wakeham, 1987), TEX86 (Schouten et al., 2006) or the

Mg/Ca of the foraminifera themselves (Elderfield and Ganssen, 2000; Mashiotta et al., 1999). Accuracy and precision of such reconstructions is debated because propagation of errors from combined inaccuracies of the analyses and the uncertainties in calibrations due to combining several proxies, is difficult to assess and seems too large for meaningful reconstructions of changes in salinity over time (Rohling, 2007). Because of the lack of a suitable alternative approach, the use of Mg/Ca to determine the temperature effect of foraminiferal $\delta^{18}O$ is continued to be applied in settings which are prone to large changes

in salinity such as the Mediterranean Sea. This calls for an independent in-situ calibration in which all the involved parameters are measured and not determined by proxy-relationships.

Culture experiments using the benthic, symbiont-barren *Ammonia tepida* (Wit et al., 2013) and the planktonic *Globigerinoides ruber* (pink) (Allen et al., 2016), showed that Na incorporation in foraminiferal shell carbonate is positively correlated with sea water salinity. A field calibration confirmed this positive correlation for the planktonic foraminiferal species *G. ruber*

(white) (Mezger et al., 2016), as well as for *Trilobatus sacculifer* (previously called *Globigerinoides sacculifer*) in the Red Sea and the Atlantic Ocean (Bertlich et al., 2018; Mezger et al., 2016). Comparison of Na/Ca-salinity calibrations shows, however, that absolute Na/Ca values and also sensitivities to salinity vary between species (Mezger et al., 2016).

When using field calibrations to constrain accuracy and precision of potential reconstruction approaches, it is important to also consider the potential impact of lateral transport of foraminifera due to (ocean) currents. Foraminifera collected at a specific

sampling location might actually have added the majority of their shell carbonate at a different location and hence under



different environmental conditions as they have been transported to the sampling location. This may add to the uncertainty in the variable to cross-correlate against or even introduce a bias in the resulting calibration. Recently this has been shown for dinoflagellate cysts (Nooteboom et al., 2019) and planktonic foraminifera, collected from sediment (van Sebille et al., 2015) and also from sediment traps (Steinhardt et al., 2014), but can also be applied to specimens collected living from the sea

surface.

Here we used a plankton pump and sea water samples collected from the Mediterranean Sea in January and February of 2016 to test viability of deconvolving salinity from combined temperature and sea water oxygen isotope reconstructions. We also investigate the potential of the newly developed salinity proxy Na/Ca in the Mediterranean Sea. Using samples collected in winter we also extent the calibration of Mg/Ca to sea water temperature for *G. ruber* towards its lower temperature tolerance

limits (14°C; Bijma et al., 1990), which is essential for the application of this species for past seasonality reconstructions.

## 2 Materials and Methods

During two cruises (NESSC Cruises 64PE406 and 64PE407, RV Pelagia) between January 12th and February 25th in 2016, a total of 98 plankton samples were collected from the surface waters of the Mediterranean Sea along an east-west transect using a plankton pump system (Ottens, 1992). Surface water was continuously pumped on board from 5m water depth and lead

through a plankton net with 100 μm mesh size. Replacing the cod-end every 6h (filtering $57m^3$ of sea water on average, constantly monitored using a water gauge), accumulated samples were washed out of the net into a 90 μm sieve, rinsed thoroughly with deionized water to remove smaller particles as well as salts, and subsequently stored onboard at -80°C. At NIOZ all plankton samples were then freeze-dried, and dry oxidation by low temperature ashing (100°C) was used to combust the organic material while minimizing potential impacts on carbonate trace metal concentrations and $\delta^{18}O$ (Fallet et al., 2009).

After ashing, samples were rinsed again thoroughly with de-ionized water and ethanol to remove potential ash residues. A variety of samples containing specimens of *G. ruber* (white) was selected to cover a large range in salinities and temperatures. Specimens used for analyses were *G. ruber* sensu stricto (Morphotype A; Kontakiotis et al., 2017) selected from the size fraction 150 - 250 μm. Surface seawater samples for stable oxygen isotopes were collected every 60 minutes from the same pump, resulting in a set of 309 samples. A volume of 2 ml was stored without headspace at 4°C during the cruise to be analyzed

at the home laboratory.

The elemental ratio of the final foraminiferal chamber (named the F-chamber) of individual shells were measured by laser ablation quadrupole inductively coupled plasma mass spectrometry (LA-Q-ICP-MS) using a circular spot with a diameter of 60-80 μm, depending on the size of the last chamber. The laser system (NWR193UC, New Wave Research) at Royal NIOZ was used in combination with a two-volume sample cell (TV2), which allows detecting variability in elemental ratios within

the foraminiferal chamber wall due to a short wash-out time of 1.8s (van Dijk et al., 2017). Ablating only F-chambers minimizes sampling of older carbonate that might have formed under different environmental conditions due to lateral and vertical transport. All specimens were ablated with an energy density of 1±0.1 J/cm2 and a repetition rate of 6Hz in a helium environment. A 0.7L/m helium flow transported the resulting aerosol to an in-house-built smoothing device before entering





the quadrupole ICP-MS (iCAP-Q, Thermo Fisher Scientific). Masses 7Li, 11B, 23Na, 24Mg, 25Mg, 27Al, 43Ca, 44Ca, 57Fe,
88Sr, 137Ba and 238U were monitored, 44Ca served as an internal standard for quantification of the associated elements. The
synthetic carbonate standard MACS-3 was used for calibration, in addition carbonate standards JCp-1, JCt-1, NFHS1 (NIOZ
foraminifera house standard; Mezger et al., 2016) as well as glass standards SRM NIST610 and NIST612 were used for
monitoring data quality. Accuracy of the analyses was 97%, while precision was 3.0% for Mg and 2.4% for Na measurements.
Stable oxygen and carbon isotopes of foraminiferal calcite were measured using an automated carbonate device (Thermo Kiel
IV) which was connected to Thermo Finnigan MAT 253 Dual Inlet Isotope Ratio Mass Spectrometer (IRMS). The NBS 19
limestone was used as a calibration standard, the NFHS1 standard was used for drift detection and correction. The standard
deviation and offset of the NBS19 and the NFHS-1 were always within 0.1‰ for $\delta^{18}O$.

Due to the large amount of material required (20 to 40 µg) and the small amount of specimens present in the samples, specimens
from different samples sometimes needed to be combined. This resulted, for example, in combining 12 and 8 µg of foraminifera
from two adjacent transects and hence, the average temperature, salinity and $\delta^{18}O_{seawater}$ for these transects was calculated
based on the relative contribution of the foraminiferal weight of the individual transects (i.e. 60 and 40 % respectively). Sea
water oxygen and hydrogen stable isotopes were analysed with the Liquid Water Isotope Analyser (LWIA; Los Gatos Research
Model 912-0008). This system measures the water samples using Off-Axis Integrated-Cavity Output Spectroscopy (OA-
ICOS). The LWIA was connected with a GC PAL from CTC Analytics to inject 1µl per measurement. To achieve this, the
GCPAL was equipped with a 1.2 µl Hamilton syringe. In-House standards (S35, S45, NSW, LGR5 and double distilled water)
were calibrated against VSMOW2-, VSLAP2- and GISP2- standard water obtained from IAEA in Vienna, using the same
setup. The use of standard water VSMOW2, which has $\delta^{18}O$ values identical to the older SMOW standard, allows for simple
comparison with older data that was calibrated using SMOW, without additional corrections. Every sample and standard was
measured 14 times sequentially, the first four runs were only used to flush the system while the last 10 measurements were
used for the analysis.  Additionally, between every sample or standard, the sample introduction line was rinsed with double
distilled water. Data were processed with LGR LWIA Post Processor Software v. 3.0.0.88. Average standard deviation per
sample was 0.14‰ for oxygen isotope measurements and 0.71‰ for hydrogen isotope measurements.

The likely provenance of the foraminifera sampled was computed by backtracking virtual particles in a high-resolution ocean
model. For this, we used the Copernicus Marine Environmental Monitoring Service (CMEMS) Global Reanalysis model. The
ocean surface currents, temperature and salinity are available at daily resolution and 1/12 degree horizontal resolution. In these
fields, we backtracked particles using the OceanParcels v2.1.1 software (Delandmeter and van Sebille, 2019; Lange and van
Sebille, 2017). We released 10,000 particles equally spaced between the start and end locations of 25 of the transects (i.e. all
for which there were sufficient foraminiferal specimens for isotope analysis), on the day these transects were sampled, and
tracked the particles back for 30 days with a 4th order Runge-Kutta algorithm with a 1 hour time step, storing local temperature,
salinity and location for each particle every day. To avoid beaching of particles, we used an unbeaching Kernel similar to that
in    Delandmeter    and    van    Sebille    (2019).    The    full    code    of    the    simulations    is    available    at
https://github.com/OceanParcels/MedForams_Daemmer/.





## 3 Results

### 3.1 The Mediterranean Sea

The sampled East-West transect spans a salinity gradient from 39.2 to 36.2 and an accompanying temperature gradient from 19°C (east) to 14°C (west). The 6 hour-intervals represented on average a distance of 57 kilometres (min 0 km, max 117 km). On average, this resulted in an internal variability of 0.14 salinity units and 0.33°C for each of the 98 transects.

Measured sea water $\delta D$ values show a range from 2.83 to 9.46‰ VSMOW in the western Mediterranean Sea and from 5.98 to 11.15‰ VSMOW in the east. Values from the individual transects were used in combination with the $\delta^{18}O_{water}$ to check for

internal consistency (Fig. 1). The $\delta^{18}O$ values of the seawater varies between 0.13 and 2.29‰ VSMOW in the west, and between 0.73 and 2.43‰ VSMOW in the east (Fig. 1). In our dataset, $\delta^{18}O$ and $\delta D$ of the water are positively correlated in both the western and eastern part of the Mediterranean Sea (Fig. 1). The sensitivities of the $\delta D$ to $\delta^{18}O$ correlations are indistinguishable. The sea water oxygen isotopes are also linearly correlated with sea water salinity (Fig. 2) and do not show an offset between the eastern and western basins (p-value < 0.001; $R^2 = 0.17$).

### 3.2 Foraminiferal geochemistry


The foraminiferal oxygen isotope ratios ($\delta^{18}O_{foraminifer}$) range from -0.41 to 0.68‰ and are significantly correlated to seawater oxygen isotope ratio (Fig. 3 a), albeit with much scatter ($R^2 = 0.42$, p-value < 0.001). The $\delta^{18}O_{foraminifer}$ are also positively correlated with sea surface salinity (Fig. 3 b) showing a similarly large amount of scatter ($R^2 = 0.44$, p-value < 0.001).

### 3.3 Na/Ca vs Salinity

Na/Ca values measured on individual F-chambers of *G. ruber* (white) from the Mediterranean Sea range from 6.8 to 12.7 mmol/mol and are positively correlated with sea surface salinity (Fig. 4 a). The variability between individuals (1-2 mmol/mol) observed within transects is orders of magnitude higher than the analytical uncertainty (RSD of 5%) and is also higher than the uncertainty in the slope of the Na/Ca-salinity calibration (Fig. 4 a).

### 3.4 Mg/Ca vs Temperature

Mg/Ca-values measured on individual F-chambers of *G. ruber* (white) from the Mediterranean Sea range from 1.34 to 7.63 mmol/mol and are positively correlated with in-situ measured sea surface temperatures, although the temperature range sampled during winter time was rather narrow (Fig. 4 b).

### 3.5 Particle backtracking

Particle backtracking shows that foraminifera collected at each transect might actually have travelled long distances within the

30days prior to sampling at the sample locations. The length of the modelled trajectories varies greatly from location to





location, ranging between 200-500km. This resulted in a variabilities (SD) within one transect ranging from 0.11 to 1.0°C and 0.03 to 0.4 salinity units.

## 4 Discussion

### 4.1 Salinity, $\delta^{18}$O and $\delta$D of the sea water

A single uniform and stable trend in sea water stable isotopes with salinity is a prerequisite for reconstructing past salinities. This is important not only when using the stable oxygen isotopes measured on foraminiferal shell carbonates, but also for the interpretation of the hydrogen isotopic composition of alkenones, which are also used as proxies for paleo-salinity (Schouten et al., 2006; Vasiliev et al., 2013; Weiss et al., 2017).

The data presented here substantially increases the amount of data on the relation between salinity and water isotopes of the 170 Mediterranean (Fig. 2). Although the new data clearly overlap with existing data, we also observe slight differences in the average salinity to $\delta^{18}$O relationship for the different data sets. The overall lower $\delta^{18}$O values of sea water measured here compared to the combined set of surface sea water isotopes from Stahl and Rinow (1973), Pierre et al. (1986), Gat et al. (1996), Pierre (1999) and Cox (2010) of approximately 0.3‰ (Fig. 2) may be explained by inter-decadal, seasonal and geographical variability between sample sets, or a combination of these factors. Importantly such offsets also give a first order indication of 175 the limit to the accuracy and precision of reconstructions of past salinity using a combined temperature-stable isotope approach from the primary relationship used.

Although Gat et al. (1996) reported a markedly different $\delta$D/$\delta^{18}$O relationship for the Eastern Mediterranean Sea compared to that of the Western Mediterranean Sea, our results show no sign of such a longitudinal discontinuity for the same area (Fig. 2). This implies that the water isotopic composition of the entire Mediterranean Sea can primarily be described by a single 180 mixing line between two end-members, with high versus lower salinity, respectively. The remarkable trend between $\delta$D/$\delta^{18}$O observed previously by Gat et al. (1996) was explained as a deuterium excess effect due to a combination of the composition of the lowermost air vapor and mixing with the enriched surface waters, most notable in winter months. The discrepancy in $\delta$D/$\delta^{18}$O relationship observed between our data and those of Gat et al. (1996) may be due to inter-decadal variability in the hydrological cycle or by differences in seasonal coverage. Potentially the observations of Gat et al. (1996) were hence either 185 related to unusual conditions or the hydrological cycle in the eastern Mediterranean has recently changed considerably. Either way the observed offset between the western and the eastern basin is apparently not stable and should therefore probably not be considered when using Mediterranean stable isotope signatures for reconstructing paleo-salinities.



### 4.2 Na/Ca vs Salinity

The Na/Ca ratios measured on the carbonate shells of *G. ruber* from the Mediterranean Sea are significantly and linearly correlated to salinity (Fig 4 a). This relationship is similar to the one reported previously for plankton pump-collected *G. ruber* from the Red Sea (Mezger et al., 2016). Mezger et al. (2016) suggested that there might a combined effect of different environmental factors such as carbonate chemistry, salinity and temperature on the Na/Ca values in the field-collected specimens. In the Red Sea it is not possible to decouple these factors as they are strongly related. Since in contrast to the Red

Sea where there is a strong negative correlation between salinity and temperature, the Mediterranean sea surface salinity and temperature are positively correlated to each other, comparing our data to that of Mezger et al. (2016) allows to decouple temperature from salinity (Fig. 5). This shows that the correlation between foraminiferal Na/Ca values and temperature observed in the Red Sea was not causal and more likely caused by salinity (Mezger et al. 2016). If temperature would have a significant effect on the Na/Ca values in *G. ruber,* we would expect different slopes and/or offsets for the Na/Ca to salinity

calibrations for the Mediterranean Sea and Red Sea. This implies that temperature has no or only a minor impact on Na/Ca ratios in *G. ruber* shells.

   The average standard deviation in Na/Ca values for a given salinity corresponds approximately to 2 salinity units, using the calibration given here (Fig. 4 a). This large variability is similar to the inter-chamber and inter-specimen variability in other El/Ca ratios, such as for example in in Mg/Ca reported by Sadekov et al. (2008) and appears to be inherent to single-chamber

El/Ca (de Nooijer et al., 2014b). It has been suggested that such variability between individuals and also between different chambers of the same individual, may be caused by differences in living depth (and hence environmental conditions (Mezger et al., 2018)), lateral transport (van Sebille et al., 2015) or variability in element incorporation during biomineralization due to vital effects (Erez, 2003; de Nooijer et al., 2014a; Spero et al., 2015) or individual timing of chamber formation (Dämmer et al., 2019). The relatively large scatter in Na/Ca values observed for single chambers (Fig. 4 a) implies that accurate and precise

reconstructions of salinity can only be based on combining a substantial number of specimens (Wit et al., 2013).

### 4.3 *G. ruber* Mg/Ca values

   The increase in Mg/Ca in *G. ruber* with temperature (Fig. 4 b) fits recent calibration efforts for Mg-incorporation and temperature (e.g. Gray et al., 2018). Since salinity and inorganic carbon chemistry also both affect Mg incorporation in this

species (Gray et al., 2018; Kisakürek et al., 2008), and the Mediterranean exhibits large gradients in these parameters, it is necessary to correct measured Mg/Ca values for these parameters. After normalizing Mg/Ca values to a sea water salinity of 35, using the calibration of Gray et al. (2018), the dependency of the Mg/Ca on temperature is similar to previously reported calibrations (e.g. Gray et al., 2018), although the Mg/Ca values at the lower most temperatures tend to be somewhat higher than expected (Fig. 8).





Adding our results to published Mg/Ca-temperature-calibrations for *G. ruber* (Anand et al., 2003; Babila et al., 2014; Fallet et al., 2010; Friedrich et al., 2012; Gray et al., 2018; Haarmann et al., 2011; Huang et al., 2008; Kisakürek et al., 2008; Mathien-Blard and Bassinot, 2009; McConnell and Thunell, 2005; Mohtadi et al., 2009) now extends the combined calibration to lower temperatures (i.e. < 18°C), maintaining the same comparatively low temperature sensitivity in the colder part of the calibration (Fig. 6). This not only increases confidence in the application of Mg/Ca in this species as a paleotemperature reconstruction

tool for colder temperatures, but also support application of individual foraminiferal Mg/Ca values for reconstructing seasonality (Wit et al., 2010). Although low densities were reported previously for G. ruber in the Mediterranean Sea during winter time, including being absent in large areas (Bárcena et al., 2004; Pujol and Grazzini, 1995) our finding implies that lowest values in Mg/Ca can be related to winter temperatures. *G. ruber* is not only present throughout the year, but it also registers the in-situ temperature, also during seasons which are close to its lower temperature limit. Admittedly the large scatter

also observed at one single sampling time (i.e. season) makes the deconvolution of seasonality from analyzing single specimen Mg/Ca values challenging.

## 4.4 δ¹⁸O_foraminifer + particle backtracking

### 4.4.1 Particle backtracking

Horizontal transport of planktonic foraminifera may increase exposure to variable environmental conditions, including different temperatures, salinities and seawater stable isotope compositions (van Sebille et al., 2015). Comparing the sampled transects with the calculated back tracking trajectories shows that especially close to the straits (Alboran Sea and Strait of Sicily) the area where the foraminifera might be derived from, potentially extends over considerable distances and therefore, variability in environmental parameters. With the surface variability in temperature and salinity during the sampling period,

the calculated variability in these parameters varied between ± (1SD) 0.11 and 1.03°C per transect and 0.04 and 0.39 salinity units per transect (Fig. 7 b, c). This means that the majority of foraminifera experienced a variability of approximately 0.5°C and 0.15 salinity units.

When considering calibrations, this is not affecting the measured proxy variables as the as the difference may be unbiased, but adds to the uncertainty of the environmental parameter to be reconstructed. Since foraminifera grow by periodically adding

chambers and since the size of the added chambers increases exponentially in many species, the carbonate added closer to the sampling location makes up a larger proportion of the total shell mass than carbonate added at earlier life stages. This implies that although the first chambers mostly formed further away from the sampling location, this has a relatively minor impact on average shell composition and hence the calibration. Therefore, the plotted back tracking trajectories (Fig. 7 a-c) indicate the largest possible range of conditions experienced by a single foraminifer. This is relevant when considering whole-shell

chemistry (i.e. oxygen isotopes; Fig. 4 a) and to a lesser extent also when considering the elemental composition of the final



chamber (Fig. 4 b). The last chamber is affected by a much smaller range in environmental conditions, i.e. only the timespan during which the final chamber was built, not more than a few days prior to sampling.

### 4.4.2 Impact on $\delta^{18}O_{foraminifer}$

Since $\delta^{18}O$ of the calcite could not be measured on F-chambers only, like for element/Ca ratios, and several specimens were needed for a single analysis, results reflect average composition of foraminiferal populations at the sampling areas. The averaging effectively cancels out differences due to inter- and intra-individual variability, but not offsets due to lateral transport. When transport directions are largely uniform, this result in biases and should not add to the scatter in the calcite's isotope composition. Hence this transport affects the calibration, but does not affect precision.

### 4.4.3 Implications for proxies

Combining existing calibrations for foraminiferal Mg/Ca and temperature (Gray et al., 2018) and calibrations relating $\delta^{18}O_{foraminifera}$ with temperature (Mulitza et al., 2003), the $\delta^{18}O_{seawater}$ can be calculated. With our dataset we here assess the quality of such reconstructions by comparison to measured $\delta^{18}O_{seawater}$ (Fig. 8). Calculated and measured $\delta^{18}O_{seawater}$ do not only not follow a 1:1 correspondence, but are not correlated at all (p-value > 0.05) which is likely due to uncertainties in the

different proxy calibrations, analytical uncertainties, heterogeneous element and isotope composition within and between specimens, and variability in the location and timing of their calcification. The lack of a correlation between calculated and measured $\delta^{18}O_{seawater}$ in our dataset implies that calculating salinity from reconstructed $\delta^{18}O_{seawater}$ values will not yield meaningful salinity reconstructions, since reconstructed values for $\delta^{18}O_{seawater}$ are not correlated to in situ measured $\delta^{18}O_{seawater}$. Calculating salinities from $\delta^{18}O_{seawater}$ clearly adds much uncertainty due to spatial and temporal variability in the correlation

of these two parameters (Conroy et al., 2017; LeGrande and Schmidt, 2006; McConnell et al., 2009).

In our dataset, the uncertainty in salinity estimates based on $\delta^{18}O_{seawater}$ is much smaller when using in situ measured temperatures (Fig. 8). This shows that the uncertainty or offset in temperatures derived from Mg/Ca, even though the Mg/Ca-temperature relationship is studied relatively extensively for *G. ruber*, is most likely the most limiting step. Even though in our dataset temperatures reconstructed from Mg/Ca deviated less than 2°C from the measured temperature, these small offsets

have a large effect on the reconstructed $\delta^{18}O_{seawater}$.

Combining all foraminiferal shell chemistry results show that salinities based on $\delta^{18}O$ and Mg/Ca may under some specific conditions allow calculating past salinity, but the uncertainties in $\delta^{18}O_{seawater}$ are large even in a setting with a large salinity gradient such as the Mediterranean. This is in line with predictions of uncertainty based on theoretical considerations (Rohling, 2007). The limiting step in these calculations is the reconstruction of past temperatures, which should be better than 2 degrees.

The development, validation and improvement of other, more direct salinity proxies such as foraminiferal Na/Ca therefore remains crucial for more reliable paleo-salinity reconstructions.



## 5. Conclusion

Using plankton pump samples from the Mediterranean Sea, we showed that 1) the relationship of Mg/Ca in *G. ruber* and sea
water temperature at lower temperatures follows an exponential relationship, therefore the proxy can now also be applied to
lower temperature ranges (<18°C) than before, covering almost the entire temperature tolerance range of that species, though
sensitivity of the calibration is comparatively low at low temperatures, 2) the combination of foraminiferal $\delta^{18}O$ and Mg/Ca
together with assumptions about $\delta^{18}O_{seawater}$ values and $\delta^{18}O_{seawater}$ – salinity relationships does not lead to useful
reconstructions of seawater salinity 3) foraminiferal Na/Ca correlates well with sea surface salinity and is independent from
temperature, making it a potentially valuable tool for salinity reconstructions.

**Data availability**: Upon publication, the data on which this manuscript is based will be available at the 4TU.Centre for
Research Data (data.4tu.nl/repository).

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





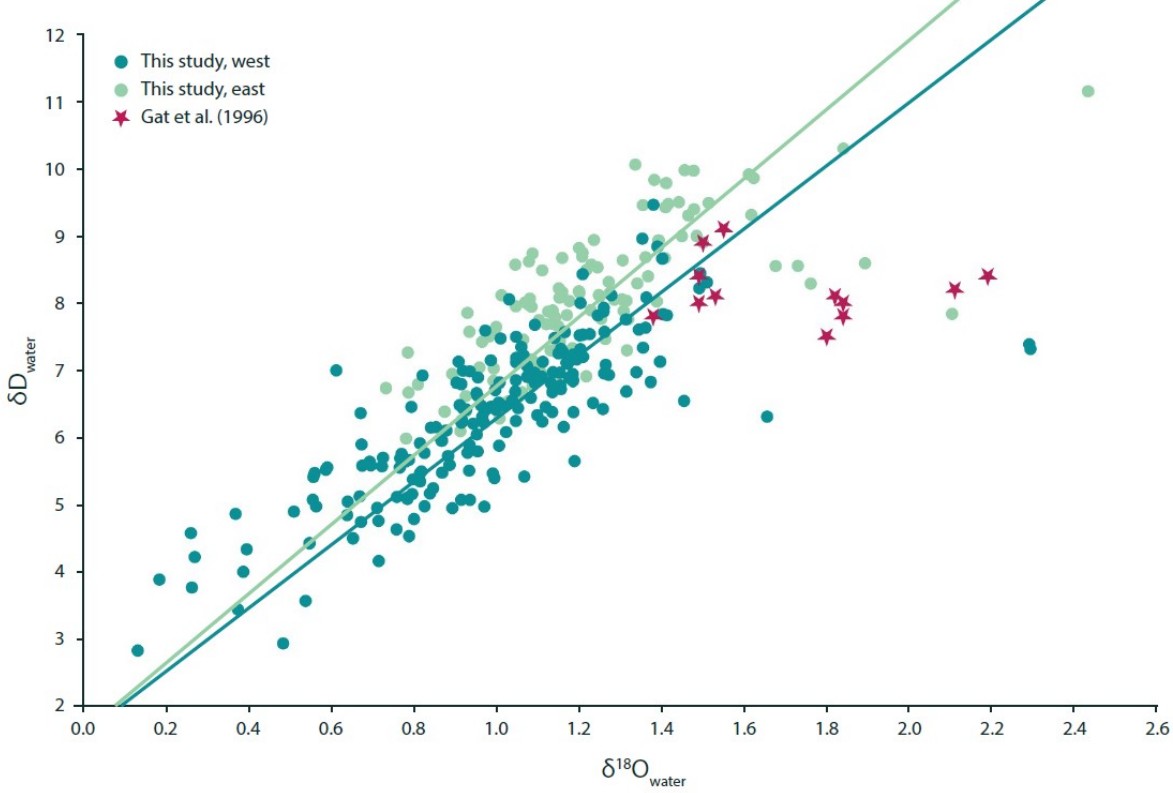

**Figure 1: The δD of the Mediterranean surface sea water is positively correlated with the local δ$^{18}$O. The orthogonal regression of the western Mediterranean can be described as δD$_{water}$ =4.82*δ$^{18}$O$_{water}$+1.67 (dark green). The eastern Mediterranean is very similar to the western basin, the relationship between sea water δ$^{18}$O and δD is δD$_{water}$ =5.19* δ$^{18}$O$_{water}$+1.68 (light green) here. In both areas the relationship is very different from the observations made by Gat et al. (1996).**


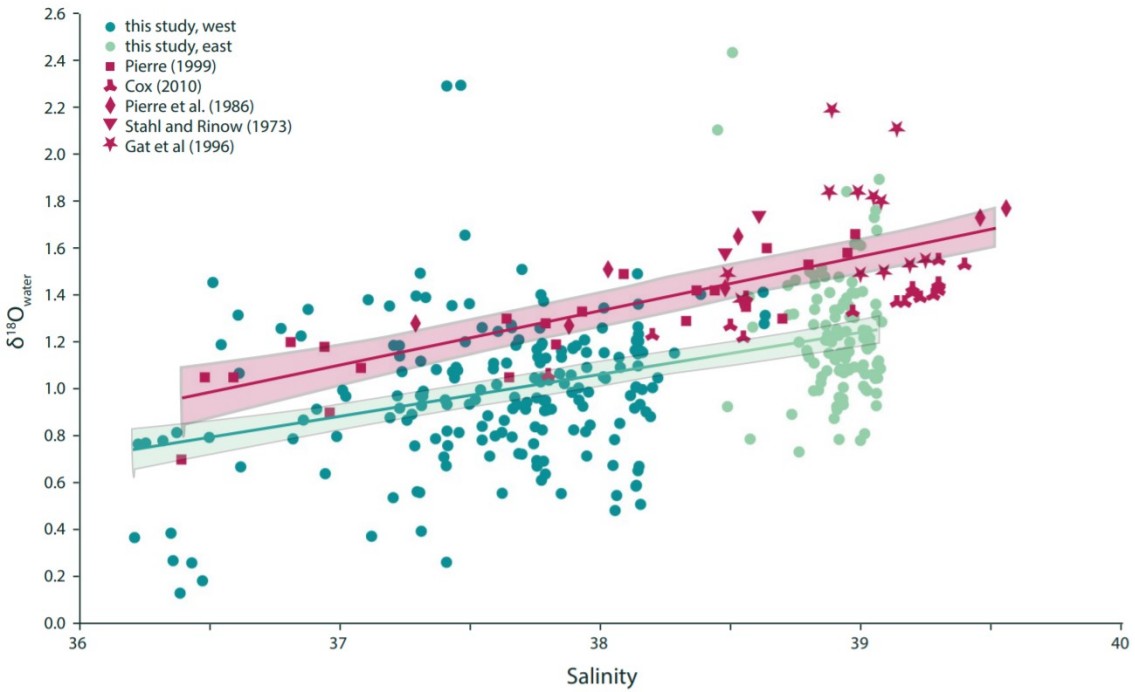

**Figure 2: Surface sea water δ18O is positively correlated with sea surface salinity in the Mediterranean Sea, the relationship observed can be described as δ18Owater=0.17*S-5.39 (p-value < 0.001). Previously published data can be combined into one dataset with a similar relationship with a slightly steeper slope, that is offset towards relatively higher δ18O (δ18Owater=0.22*S-7.19; p-value < 0.001).**


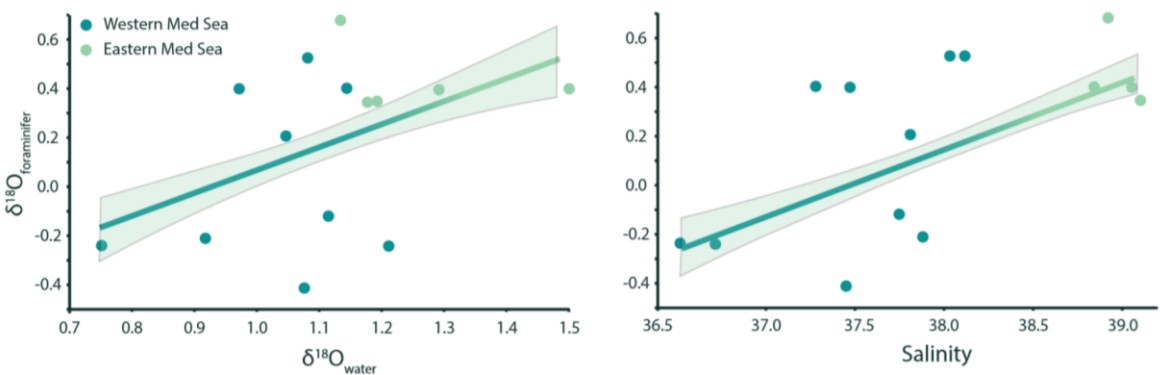

**Figure 3:  G. ruber δ18O measurements are positively correlated (p-value < 0.001) to both sea water δ18O (a) and salinity (b). The relationships can be described using the following equations: δ18Oforaminifera=0.28*S-10.59 and δ18Oforaminifera =0.95* δ18Owater–0.89.**





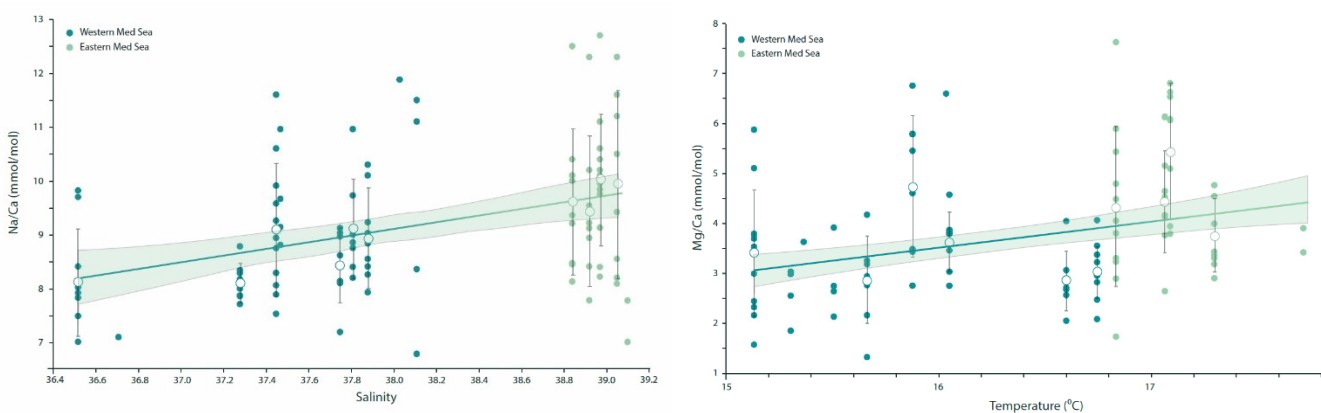


**Figure 4: (a)** Na/Ca measured in G. ruber F-chambers collected as living specimens from the eastern and western Mediterranean Sea correlates well with local salinity (p-value < 0.001, Na/Ca=0.66 * S-16.07), even though a large natural spread of elemental composition around the mean values per station exists. For salinities with more than 5 individual Na/Ca measurements, hollow circles with whiskers indicate average values and standard deviations. **(b)** Mg/Ca in F-chambers of G. ruber specimens collected

from the water column of the Mediterranean Sea is positively correlated with sea surface temperature and can be described with the linear relationship Mg/Ca=0.47*T-3.98 (p-value < 0.05). For temperatures with more than 5 individual Mg/Ca measurements, hollow circles with whiskers indicate average values and standard deviations.

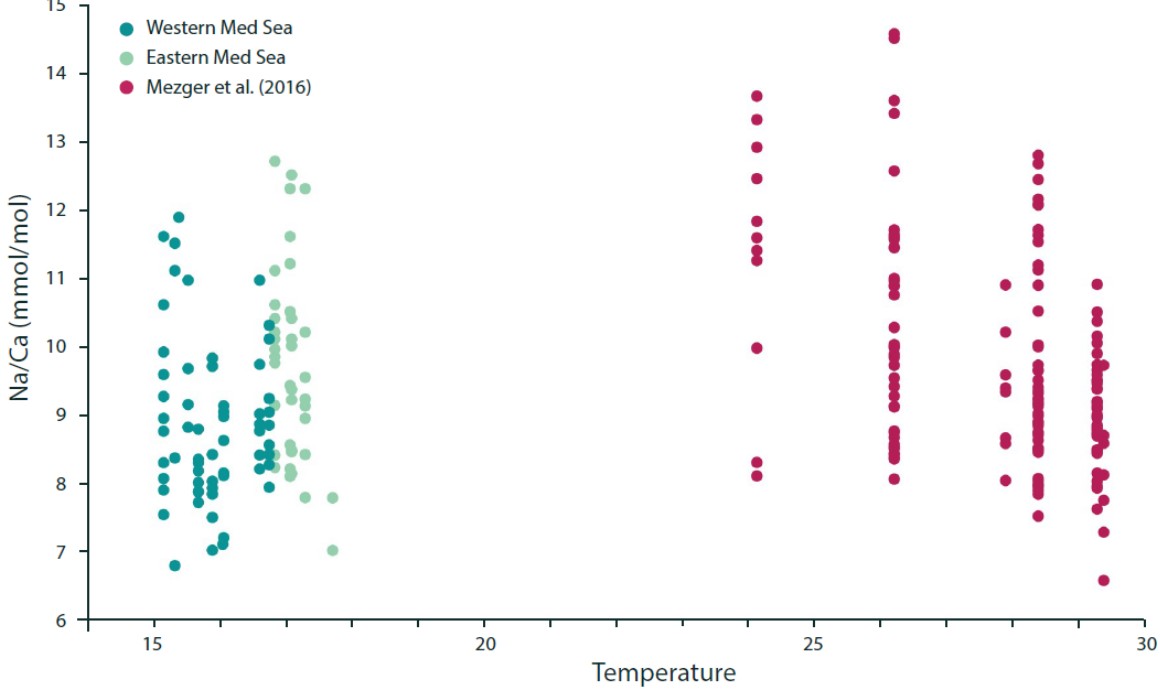


**Figure 5:** The ratio of Na/Ca in G. ruber appears to be independent from sea water temperature. While Mezger et al. (2016) showed a negative relationship between temperature and foraminiferal Na/Ca in specimens collected from the Red Sea, the addition of new data from the Mediterranean Sea shows clearly that the previously hypothesized negative impact of temperature on Na/Ca is likely





an artefact of the negative relationship of temperature and salinity in the Red Sea and that temperature has no significant impact on Na/Ca.

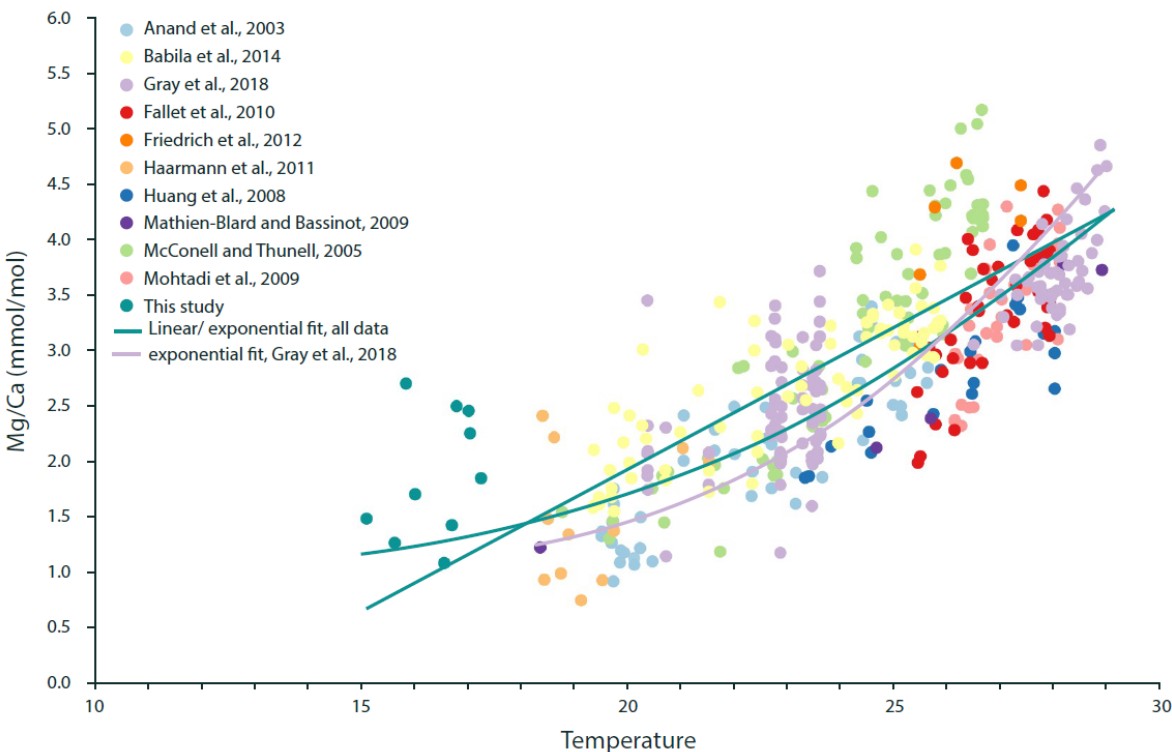

**Figure 6: The relationship between Mg/Ca in G. ruber and temperature during calcification can be described using the following exponential equation: Mg/Ca=0.278*exp(0.093*T) for a temperature range from 15.1 to 29.1°C.**






**Figure 7: (a) Example of back-tracked pathways for a single transect (the one marked by a white rectangle in panel c). The colour indicates the time before sampling up to 30 days. (b) Analysing the different environmental conditions at the different locations of these potential paths show that foraminifera sampled very likely experienced a large range in temperatures as well as salinities. (c) The variability in potentially experienced environmental conditions varies considerably from location to location, as indicated by notation of 1 standard deviation for both parameters for each sampling location. Maps in (a) and (c) were generated using Ocean Data View version 4 (Schlitzer, 2018).**







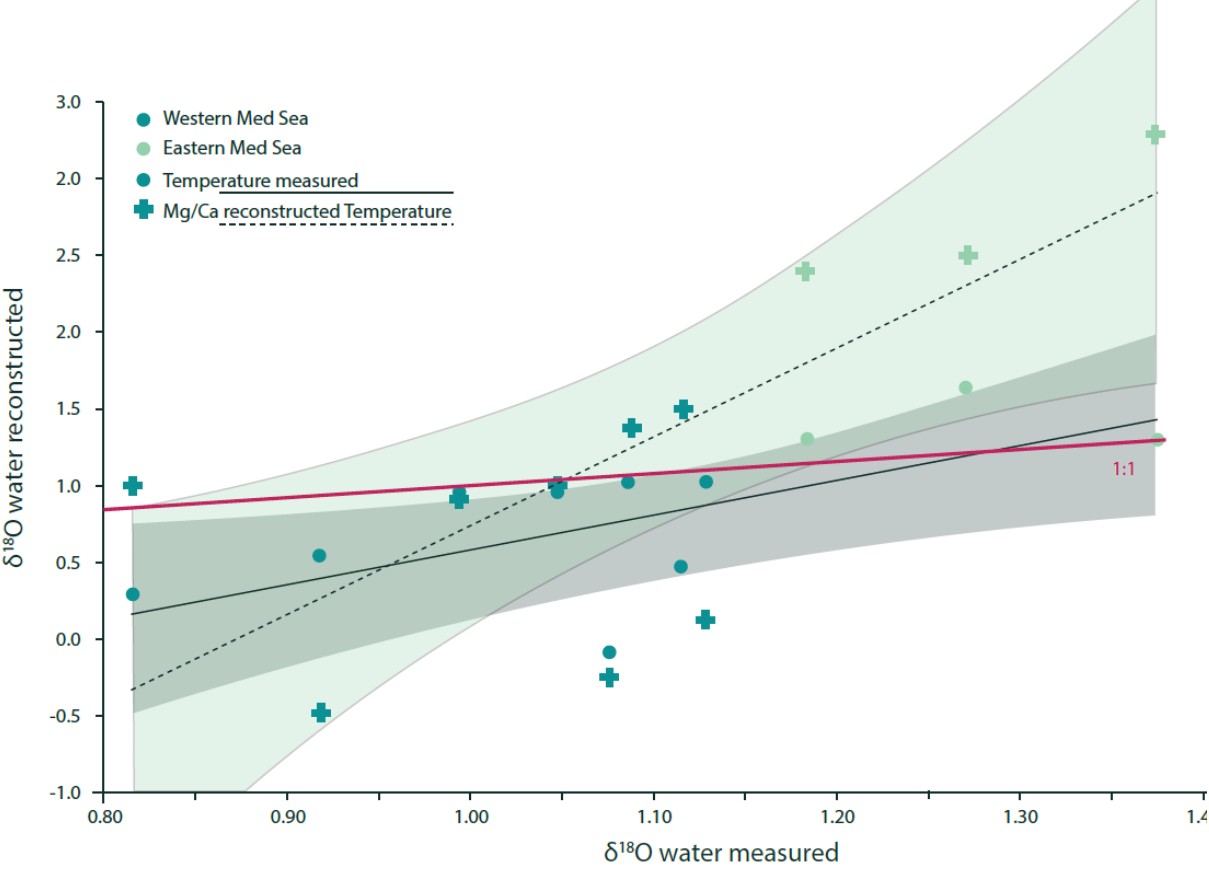

**Figure 8: The relationship between δ18Oseawater measured in the Mediterranean Sea and δ18Oseawater calculated from foraminiferal geochemistry (*G. ruber* white). The relationship shown with dashed lines and cross shaped markers represents values calculated using foraminiferal δ18O as well as Mg/Ca as an additional temperature proxy to decouple the effect of temperature and salinity on δ18O. This relationship is non-significant (p-value >0.05). The relationship shown with the continuous lines and circular markers shows the same samples, but instead of using temperature values derived from foraminiferal Mg/Ca ratios, in situ measurements for temperature were used, the relationship can be described as $\delta^{18}O_{water\_reconstructed}=2.62(\pm0.69)*\delta^{18}O_{water\_measured}-63.99(\pm26.11)$. The temperature gradient was 2.2°C.**