# Peer review of "Evaluation of oxygen isotopes and trace elements in planktonic foraminifera from the Mediterranean Sea as recorders of seawater oxygen isotopes and salinity"

_Climate of the Past, 2020_

## Referee Comment (RC1) · Anonymous Referee #1 · 8 Apr 2020

General comments:

The manuscript entitled "Evaluation of isotopes and elements in planktonic foraminifera from the Mediterranean Sea as recorders of seawater oxygen isotopes and salinity" by Dämmer L.K, and coauthors provide new data about the marine proxy calibrations.

The study is based on planktonic foraminifera and sea water samples collected from the Mediterranean Sea in January and February of 2016. The investigation focus on the relation between element/Ca ratios, stable oxygen isotopes of the foraminiferal species

Globigerinoides ruber (alba) and surface seawater salinity, isotopic composition and temperature. This is an important issue in the paleoceanographic investigations. Infact, in order to accurately interpret past climate and environment, it is fundamental to have reliable proxies.

Specific comments

It would be effective to insert in the Discussion section a short paragraph with "recommendation for the applications of the proxies" that the authors (based on this study) consider relevant for the paleo-reconstructions (i.e, using more specimens for the analyses, uncertainty in salinity estimates?, how collect the samples, etc. . ..).

In fig 2 ($\delta$18O seawater versus salinity) all data are from Mediterranean sea except data from Cox (2010) that are from North Atlantic. I do not understand why the authors use the North Atlantic data, otherwise the authors can discuss this in paragraph 4.1 (when they report geographical variability, lines 173,174).

Technical corrections

3.1 I suggest as title: seawater geochemistry or Mediterranean Sea geochemistry 4.2 The analyses were performed on Na/Ca ratios measured on the carbonate shells of G. ruber. It is G. ruber (white) as reported in the paragraph 3.3? The same observation for paragraph 4.3 Line 191- Fig. 4a (add a point) Line 226 G.ruber in italic font Fig. 3, 4, 5: G. ruber in italic font

---

## Referee Comment (RC2) · Michal Kucera (Referee) · 28 Jun 2020

The authors present a comprehensive regional dataset of trace element and stable oxygen isotope data measured on foraminifera collected from plankton samples with rich contextual physical and chemical data. The analyses were carried out to test to what degree the strong salinity (or seawater oxygen isotope) gradient in the Mediterranean could have been reconstructed from shell chemistry. The results are sobering, which I believe is not to be taken negatively but as an extremely important result, confirming the growing body of evidence that there is something we fundamentally do not

understand about the way the proxy signal in the sediment is generated. In this way, the manuscript makes an important fresh contribution to the field and the data and analyses in my opinion warrant publication in Climate of the Past.

That said, I would advise the authors to place less emphasis on the aspect of lateral advections, for reasons explained below, and to provide a more explicit quantitative evaluation of the magnitude and direction of the various candidate processes invoked to explain the large scatter. Beyond the individual comments listed below, I would like the authors to explain if/how they dealt with the carbonate ion effect on all of the proxies (oxygen isotopes and Mg/Ca in particular), as this is not really clear from the text and I would like to draw to their attention the possibility that the analyses of the G. ruber from the plankton in the chosen size fraction could have been affected by differential contribution of specimens representing pre-adult G. elongatus, which may follow a different calibration line. Perhaps the results already contain some hints (bimodality or not of the single-shell measurements, for example)?

Finally, I would like to urge the authors to make sure that the data that will be make available on the Utrecht data server are as comprehensive as possible and that they are stored in a way that they will be found in any future attempts to synthesize seawater or foraminifera chemistry data.

Taken together, these points and the individual points below all aim to make the most out of the nice dataset that the authors have, which I believe they will be able to do without having to substantially restructure the paper or change its conclusions.

Comments to individuals points:

Title: Instead of "isotopes and elements", I would recommend to be either more specific (oxygen isotopes and trace elements) or less specific (shell geochemistry), or else the title appears to promise more than what is delivered.

Line 30: large scale

Line 54: continues to be

Line 66: please specify what exactly "has been shown for foraminifera". In my opinion the effect of expatriation on shell chemistry in foraminifera has been previously shown by the work of Ganssen and Kroon in the Red Sea, but not really outside of that extreme environment. The studies cited in this place were mainly concerned with attempts to use particle tracking in models and describe potential effects, rather than documenting these effects empirically, or the empirical detection was indirect, inferred from sediment trap material where the dwelling depth is unknown.

Line 87: there are no formally and objectively defined and biologically or ecologically meaningful morphotypes within the species G. ruber. The concept of "morphotypes", re-introduced into the literature by Wang, has been superseded by the discovery based on genetic data (Aurahs et al., 2011), that the species concept as introduced by Parker (1965) is incorrect and that the species G. elongatus, synonymised by her with G. ruber, should have been retained. The same genetic data have also revealed that the pink and white varieties of G. ruber are genetically distinct and these have been now formally distinguished at the level of subspecies. The correct label of the analysed taxon is thus Globigerinoides ruber albus (Morard et al., 2020), with morphology corresponding to what Kontakiotis et al. (2017) label as Morphotype A.

Line 88: I fully understand the decision to concentrate on the relatively small size fraction for analyses, as this likely yielded most material. However, I would like to point out that Aurahs et al. (2011), also working with plankton material, also from the Mediterranean, showed that the features distinguishing G. ruber albus from G. elongatus are not yet present among all specimens in the plankton, allowing separation of plankton-derived specimens to the ruber and elongatus only to about 75 % accuracy. Since G. elongatus is abundant (if not dominant) in the Mediterranean, the authors must consider the possibility that some of the analysed specimens may have belonged to that species.

Line 104: The methods section here is not entirely clear in how the oxygen isotopes were measured. Whereas it is clear that Mg/Ca was determined on final chambers of individual shells, the authors should specify if the isotopes were also measured on final chambers or whole shells, on single shells or multiple shells (and then how many) and whether the same shells as for Mg/Ca were used or different shells. This has all implications for the understanding of the origin of the apparent noise in the measurements.

Figure 1: I agree that the two regressions (correctly using a total least squares approach) are similar, but could the authors please provide a formal statistical test for the similarity of the slopes, to support their statement that the sensitivities are indistinguishable, and for the equality of the intercepts, to dispel the impression that the regression lines are offset, indicating different endmember composition? Also, I am not convinced that it is correct to consider the results of Gat et al. (1996) as being different, as all of their values fall within the range of the presented data.

Figure 2: Could the authors please state which regression has been used here and also provide a formal test for the lack of difference in the east and west and for the presence of a difference in the slope and intercept between their data and literature data? Also please provide R2 for all regressions in the figure caption and/or text.

Line 145: Considering that seawater oxygen isotopes and salinity only correlated with R2 od 0.2, the authors need an explanation for what the isotopes in foraminifera correlated more strongly with both variables. Could it be that each of the variables explains a different part of the total variance? Then, a multiple regression of foraminifera isotopes against seawater isotopes an salinity should explain significantly more variance. If it does not, it means that the two explanatory variables explain the same amount of variance. This could be because of a fortuitous choice of sampling and the authors should thus also calculate the R2 for salinity and seawater isotopes only for the samples shown in Figure 3.

Figure 4: Could the authors again specify what regression has been used and how exactly the regression lines were calculated (regression of individual values or of the means)? Please state R2 for all regressions. Also, the Mg/Ca to T relationship is known to be exponential, so why not fitting an exponential curve? The linearity of the relationship could simply reflect the fact that the regression is fitted over a relatively narrow temperature range.

Line 155: Considering that Mg/Ca is also changing as a function of salinity, why not plotting Mg/Ca against salinity and analyzing the strength of that relationship as well?

Line 160: it is true that the foraminifera may have travelled a long distance over the 30 days of the simulation, but I question the significance of the so derived variability for the interpretation of the shell geochemistry. Culturing observations indicate that G. ruber in the size range as analysed here produces a new chamber about every two days. Thus, the particle tracking result has no bearing on the laser-ablation data. For the isotope data, if we assume a total lifespan of 4 weeks and a life expectancy of the specimens in the analysed size range of two weeks, then the collected specimens would have only had two weeks to grow, not 30 days. On top of that, because of the exponential growth of the shell, almost all of the analysed calcite and thus almost all of the isotopic signal is present in the last few chambers of the shell, so it reality, the backtracking relevant to the analysed signal should not have been carried back for more than a week. This is not to say that the result stated here is wrong – it is just that the result is not relevant for the interpretation of the measured geochemical signals. I note that your discussion in 4.4.1 resonates well with what I write, but then I do not really understand what was the merit or the justification of showing the particle backracking results in figure 7 over 30 days?

Line 170 and onwards: please see the comments above as to the necessity to provide statistical tests to support the presence or absence of differences in regression shapes. Also, please consider the location of the sampling by Gat and yours: what if the apparent offset from your regression that he reports simply reflects the fact that

he sampled at locations where the relationship is unusually confounded by secondary variables and that your data would detect the same if you only had measurements at those locations? I am also concerned by the origin of the lower oxygen isotope values measured for the given salinity in your data: was the sampling method comparable between your data and those of the previous studies (collecting from the same depth)?

Line 202 and onwards: Considering all your specimens were collected from the surface and that you measured only the composition of the final chamber, would it not be logically at this place to reject some of the hypotheses that you list here? Otherwise, you would have to imply that the specimens migrate vertically tens of meters over a few days, or stay alive without adding new chambers for weeks to allow lateral transport to have an effect. So perhaps we are left with the variable biomineralisation as the only remaining candidate mechanism?

Line 216: I fear the Mg/Ca data are revealing more than what the authors imply. Firstly, since the authors have both temperature and salinity, they should derive the correction independently of Gray et al. (2018) or at least check if the relationship they obtain holds. Second, I wonder why the authors do not discuss the fact that once the salinity effect is removed, their Mg/Ca data are no longer correlated with temperature or if correlated then with a much steeper slope (at least this is what I see looking at Figure 8). Third, I do not agree with the statement that the corrected values are slightly higher than expected based on the global regression – I observe that they are all higher than predicted by the exponential regression (the linear regression in Figure 8 is in my opinion superfluous). Why is that? Could there be a salinity-temperature interaction affecting the salinity-Mg/Ca relationship? This is an important result that deserves some more thought.

Line 226 (and some figure captions): please make sure species names are always written in italics.

Line 228: an argument on the presence (production) of G. ruber in different seasons

in the Mediterranean would benefit from references to sediment trap data. There is a nice long time series from the west (Rigual-Hernandez et al., 2012) and a new dataset from the east (Avnaim-Katav et al., 2020, Deep-Sea Research) that could be used to support these statements.

Sections 4.4.1 and 4.4.2: I believe the authors could do better in providing quantitative constraints on the strength of the processes invoked to explain the large deviations in trace metals and oxygen isotopes from the theoretical calibration curves. For example, in section 4.4.2 they seem to imply that the oxygen isotope signal should be much less affected by the individual variability, but not by lateral transport. Notwithstanding of what the value of the 30-day calculation is, one should then ask: how much lateral transport would be needed at each of the locations to explain the isotopic scatter? Where would the calcification have to occur? Is the offset due to lateral transport large enough or not to be considered the main mechanism behind the scatter. Similarly, if all other other processes do not act on oxygen isotopes then the scatter in isotopes (residuals) should be less than in the Mg/Ca. Is it? I feel the authors should take the discussion further and provide at least first-order assessment of the strength and direction of the invoked processes and evaluate the plausibility of those processes in explaining the scatter.

Line 265: on the same note: why is the lack of correlation "likely" due to all those uncertainties? How big are these uncertainties exactly? The reader needs to see the values to be able to evaluate statements like on line 271, which are intuitively correct, but not really supported by any calculations. Please provide R2 and p for both regressions shown in Figure 8. Also, the method by which the oxygen isotopes in seawater have been estimated is not sufficiently documented. For example, it is not clear if and how the salinity effect on Mg/Ca has been considered.

Line 281: why do the authors not take this opportunity to compare the performance of Na/Ca and the combined isotope and Mg/Ca on the resulting salinity estimates? There is no need to end with a general statement, when the authors have all the data to carry

out the comparison.

---

## Author Comment (AC1) · 26 Jul 2020

We thank Anonymous Referee #1 for their encouraging words and helpful suggestions. We have adjusted the manuscript text and figure descriptions following these suggestions. Our replies (right aligned) to their comments (left aligned) are below.

**Comment:** General comments:

The manuscript entitled "Evaluation of isotopes and elements in planktonic foraminifera from the Mediterranean Sea as recorders of seawater oxygen isotopes and salinity" by Dämmer L.K, and coauthors provide new data about the marine proxy calibrations.

The study is based on planktonic foraminifera and sea water samples collected from the Mediterranean Sea in January and February of 2016. The investigation focus on the relation between element/Ca ratios, stable oxygen isotopes of the foraminiferal species Globigerinoides ruber (alba) and surface seawater salinity, isotopic composition and temperature. This is an important issue in the paleoceanographic investigations. Infact, in order to accurately interpret past climate and environment, it is fundamental to have reliable proxies.

Specific comments:

It would be effective to insert in the Discussion section a short paragraph with "recommendation for the applications of the proxies" that the authors (based on this study) consider relevant for the paleo-reconstructions (i.e, using more specimens for the analyses, uncertainty in salinity estimates?, how collect the samples, etc. . .).

   **Reply:** We have extended section "4.4.3 Implications for proxies" and added the suggested topics to our discussion.

**Comment:** In fig 2 (δ18O seawater versus salinity) all data are from Mediterranean sea except data from Cox (2010) that are from North Atlantic. I do not understand why the authors use the North Atlantic data, otherwise the authors can discuss this in paragraph 4.1 (when they report geographical variability, lines 173,174).

   **Reply:** We agree that mixing data from different basins should be avoided for this analysis. While the publication title for the reference "Cox (2010)" is indeed *"Stable Isotopes as Tracers for Freshwater Fluxes into the North Atlantic"*, Katharine A. Cox' publication does not only contain North Atlantic sea water isotope data, but also a number of Mediterranean Sea water isotope measurements which are presented in her Appendix D.2 *"Oxygen and Hydrogen Isotope Data analyzed at UC Davis: Table D.4: Station locations, depths, salinity data and isotopic parameters of the 2001 M51−3 water samples from Mediterranean, the 2004 JR106b water samples from Kangerdlussuaq Fjord, Denmark Strait and the 2005 D298 and 2008 D332 water samples from Cape Farewell"*, p. 138ff. From this data set we selected surface water δ¹⁸O and δD measurements from the Mediterranean Sea samples M51-3 to include in our analysis and Fig. 2 to compare with our own Mediterranean Sea data.

**Comment:** Technical corrections:

3.1 I suggest as title: seawater geochemistry or Mediterranean Sea geochemistry

**Reply:** We have adjusted the title of section 3.1 to "Mediterranean Sea geochemistry" following this suggestion.

**Comment:** 4.2 The analyses were performed on Na/Ca ratios measured on the carbonate shells of G. ruber. It is G. ruber (white) as reported in the paragraph 3.3?

**Reply:** Correct, all analysis was performed on the same species, as also described in section "2 Materials and Methods". We have updated the species name throughout the manuscript to *Globigerinoides ruber albus* to reflect the recent suggestions presented by Morard et al. (2019) and to avoid any ambiguity.

**Comment:** The same observation for paragraph 4.3

**Reply:** See previous reply.

**Comment:** Line 191- Fig. 4a (add a point)

**Reply:** We have added the missing point.

**Comment:** Line 226 G.ruber in italic font

**Reply:** We have changed the font to italic.

**Comment:** Fig. 3, 4, 5: G. ruber in italic font

**Reply:** We have changed the font to italic.

---

## Author Response (AR1)

[revised manuscript text omitted]

| # | Comments Reviewer 1 | Our replies | Changes made in manuscript |
|---|---|---|---|
| 1.01 | It would be effective to insert in the Discussion section a short paragraph with "recommendation for the applications of the proxies" that the authors (based on this study) consider relevant for the paleo-reconstructions (i.e, using more specimens for the analyses, uncertainty in salinity estimates?, how collect the samples, etc. . .). | We have extended section "4.4.3 Implications for proxies" and added the suggested topics to our discussion. | Added: "It is therefore crucial to choose temperature proxies carefully, use a large enough number of specimens for analysis, be aware about potential effects of lateral particle transport as well as other environmental parameters, and to be conscious about how errors propagate in paleoclimate reconstructions" to section 4.4.3 |
| 1.02 | In fig 2 ($\delta 18O$ seawater versus salinity) all data are from Mediterranean sea except data from Cox (2010) that are from North Atlantic. I do not understand why the authors use the North Atlantic data, otherwise the authors can discuss this in paragraph 4.1 (when they report geographical variability, lines 173,174). | We agree that mixing data from different basins should be avoided for this analysis. While the publication title for the reference "Cox (2010)" is indeed *Stable Isotopes as Tracers for Freshwater Fluxes into the North Atlantic*", Katharine A. Cox' publication does not only contain North Atlantic sea water isotope data, but also a number of Mediterranean Sea water isotope measurements which are presented in her Appendix D.2 "*Oxygen and Hydrogen Isotope Data analyzed at UC Davis: Table D.4: Station locations, depths, salinity data and isotopic parameters of the 2001 M51–3 water samples from Mediterranean, the 2004 JR106b water samples from Kangerdlussuaq Fjord, Denmark Strait and the 2005 D298 and 2008 D332 water samples from Cape Farewell*", p. 138ff. From this data set we selected surface water $\delta_{18}O$ and $\delta D$ measurements from the Mediterranean Sea samples M51-3 to include in our analysis and Fig. 2 to compare with our own Mediterranean Sea data | No changes made. |
| 1.03 | 3.1 I suggest as title: seawater geochemistry or | We have adjusted the title of section 3.1 to "Mediterranean | Changed title of section 3.1 to "Mediterranean Sea geochemistry" |

| | | | |
|---|---|---|---|
| | Mediterranean Sea geochemistry | Sea geochemistry" following this suggestion. | |
| | 4.2 The analyses were performed on Na/Ca ratios measured on the carbonate shells of G. ruber. It is G. ruber (white) as reported in the paragraph 3.3? | Correct, all analysis was performed on the same species, as also described in section "2 Materials and Methods". We have updated the species name throughout the manuscript to *Globigerinoides ruber albus* to reflect the recent suggestions presented by Morard et al. (2019) and to avoid any ambiguity. | Updated the species name throughout the manuscript to *Globigerinoides ruber albus.* |
| 1.04 | The same observation for paragraph 4.3 | See previous reply | - |
| 1.05 | Line 191- Fig. 4a (add a point) | We have added the missing point | Point added |
| 1.06 | Line 226 G.ruber in italic font | We have changed the font to italic. | Font changed |
| 1.07 | Fig. 3, 4, 5: G. ruber in italic font | We have changed the font to italic. | Font changed |
| | | | |
| | Comments Reviewer 2 | Our replies | Changes made in manuscript |
| 2.01 | The authors present a comprehensive regional dataset of trace element and stable oxygen isotope data measured on foraminifera collected from plankton samples with rich contextual physical and chemical data. The analyses were carried out to test to what degree the strong salinity (or seawater oxygen isotope) gradient in the Mediterranean could have been reconstructed from shell chemistry. The results are sobering, which I believe is not to be taken negatively but as an extremely important result, confirming the growing body of evidence that there is something we fundamentally do not understand about the way the proxy signal in the sediment is generated. In this way, the manuscript makes an important fresh contribution to the field and the data and analyses in my opinion | We will rephrase parts of the discussion to put less emphasis on the impact of lateral transport, following the suggestions of the reviewer.

We will include explanations concerning the carbonate ion effect in the next version of the manuscript.

We have included a comment about the possibility that specimens of *G. elongatus* might have incorrectly be identified as *G. ruber albus* in section 2 Materials and Methods, as described in our reply to a later comment. We have found no bimodality in the single shell LAQICPMS measurements of our specimens and therefore assume that the effect of species misidentification is minimal. | We combined 4.4.1 and 4.4.2 and rephrased the heading to reflect the decrease in emphasis on the particle backtracking results. We also shortened the text and instead now focus on the most important consequence of this exercise.

Added "The Mg/Ca values used here were not corrected for salinity or effects, since salinity is the target parameter that has to be reconstructed and is thus treated as unknown. Even though there is a carbonate ion effect on the Mg/Ca in *G. ruber albus* (Evans et al., 2016; Gray et al., 2018; Kisakürek et al., 2008), the measured values were not corrected for this, since this factor is also unknown in paleo-reconstructions." To section 4.4.3

"Specimens used for analyses were selected from the size fraction 150 - 250 µm, even though it has been reported that |

| | | | |
|---|---|---|---|
| | warrant publication in Climate of the Past. That said, I would advise the authors to place less emphasis on the aspect of lateral advections, for reasons explained below, and to provide a more explicit quantitative evaluation of the magnitude and direction of the various candidate processes invoked to explain the large scatter. Beyond the individual comments listed below, I would like the authors to explain if/how they dealt with the carbonate ion effect on all of the proxies (oxygen isotopes and Mg/Ca in particular), as this is not really clear from the text and I would like to draw to their attention the possibility that the analyses of the G. ruber from the plankton in the chosen size fraction could have been affected by differential contribution of specimens representing pre-adult G. elongatus, which may follow a different calibration line. Perhaps the results already contain some hints (bimodality or not of the single-shell measurements, for example)? | | at this size fraction *G. ruber albus* and *Globigerinoides elongatus* cannot always be confidently distinguished due to similar morphology (Aurahs et al., 2011)" |
| 2.02 | Finally, I would like to urge the authors to make sure that the data that will be make available on the Utrecht data server are as comprehensive as possible and that they are stored in a way that they will be found in any future attempts to synthesize seawater or foraminifera chemistry data. | All the environmental and geochemical data from this project will be available for download at the 4TU.Centre for Research Data where we are certain it can easily be found and accessed by anyone | No changes made. |
| 2.03 | Taken together, these points and the individual points below all aim to make the most out of the nice dataset that the authors have, which I believe they will be able to do | We have changed the manuscript title according to this suggestion into "Evaluation of oxygen isotopes and trace elements in planktonic foraminifera from | Changed title of the manuscript to "Evaluation of oxygen isotopes and trace elements in planktonic foraminifera from the Mediterranean Sea as recorders |

| | | | |
|---|---|---|---|
| | without having to substantially restructure the paper or change its conclusions. Comments to individuals points:
Title: Instead of "isotopes and elements", I would recommend to be either more specific (oxygen isotopes and trace elements) or less specific (shell geochemistry), or else the title appears to promise more than what is delivered. | the Mediterranean Sea as recorders of seawater oxygen isotopes and salinity" | of seawater oxygen isotopes and salinity" |
| 2.04 | Line 30: large scale | We have corrected this typo. | Changed from "largescale" to "large scale" |
| 2.05 | Line 54: continues to be | We have corrected this typo. | Changed from "is continued to be" to "continues to be" |
| 2.06 | Line 66: please specify what exactly "has been shown for foraminifera". In my opinion the effect of expatriation on shell chemistry in foraminifera has been previously shown by the work of Ganssen and Kroon in the Red Sea, but not really outside of that extreme environment. The studies cited in this place were mainly concerned with attempts to use particle tracking in models and describe potential effects, rather than documenting these effects empirically, or the empirical detection was indirect, inferred from sediment trap material where the dwelling depth is unknown. | We agree with this comment and thus have changed the manuscript text from "has been shown" to "has been suggested" to account for the fact that the quoted studies are not empirical. We also added the suggested reference Ganssen and Kroon (1991) to the Introduction. | Changed to "Recently this has been shown suggested for dinoflagellate cysts (Nooteboom et al., 2019) and planktonic foraminifera, collected from the water column (Ganssen and Kroon, 1991), from sediment (van Sebille et al., 2015) and also from sediment traps (Steinhardt et al., 2014), but can also be applied to specimens collected living from the sea surface." |
| 2.07 | Line 87: there are no formally and objectively defined and biologically or ecologically meaningful morphotypes within the species G. ruber. The concept of "morphotypes", re-introduced into the literature by Wang, has been superseded by the discovery based on genetic data (Aurahs et al., 2011), that the species concept as introduced by Parker (1965) is incorrect and that the species | We have adjusted this paragraph by using the correct taxon label *G. ruber albus*, including the reference Morard et al. (2019) and exclude the mention of morphotypes (see next comment for updated version of this line). We also updated the species name to *G. ruber albus* throughout the rest of the manuscript. | Changed to "A variety of samples containing specimens of *G. ruber albus* (Morard et al., 2019) was selected to cover a large range in salinities and temperatures. Specimens used for analyses were selected from the size fraction 150 - 250 μm, even though it has been reported that at this size fraction *G. ruber albus* and *Globigerinoides elongatus* cannot always be confidently distinguished due to similar |

| | | | morphology (Aurahs et al., 2011)." |
|---|---|---|---|
| | G. elongatus, synonymised by her with G. ruber, should have been retained. The same genetic data have also revealed that the pink and white varieties of G. ruber are genetically distinct and these have been now formally distinguished at the level of subspecies. The correct label of the analysed taxon is thus Globigerinoides ruber albus (Morard et al., 2020), with morphology corresponding to what Kontakiotis et al. (2017) label as Morphotype A. | | |
| 2.08 | Line 88: I fully understand the decision to concentrate on the relatively small size fraction for analyses, as this likely yielded most material. However, I would like to point out that Aurahs et al. (2011), also working with plankton material, also from the Mediterranean, showed that the features distinguishing G. ruber albus from G. elongatus are not yet present among all specimens in the plankton, allowing separation of plankton-derived specimens to the ruber and elongatus only to about 75 % accuracy. Since G. elongatus is abundant (if not dominant) in the Mediterranean, the authors must consider the possibility that some of the analysed specimens may have belonged to that species. | The decision to take specimens from size fraction 150-250µm was indeed made after an initial examination of size fraction >250µm yielded insufficient material for analysis. We would like to thank the referee for bringing the Aurahs et al. (2011) study to our attention. After reading it we decided to include a comment about this in section 2 Materials and Methods, it now reads: *"A variety of samples containing specimens of* G. ruber albus *(Morard et al. 2019) was selected to cover a large range in salinities and temperatures. Specimens used for analyses were selected from the size fraction 150 - 250 µm, even though it has been reported that at this size fraction* G. ruber albus *and* Globigerinoides elongatus *cannot always be confidently distinguished due to similar morphology (Aurahs et al., 2011)."* | Changed to: "A variety of samples containing specimens of *G. ruber albus* (Morard et al., 2019) was selected to cover a large range in salinities and temperatures. Specimens used for analyses were selected from the size fraction 150 - 250 µm, even though it has been reported that at this size fraction *G. ruber albus* and *Globigerinoides elongatus* cannot always be confidently distinguished due to similar morphology (Aurahs et al., 2011)" |
| 2.09 | Line 104: The methods section here is not entirely clear in how the oxygen isotopes were measured. Whereas it is clear that Mg/Ca was determined on final chambers of individual | We agree that the previous version of this section was not sufficiently clear on this point. We have adjusted this paragraph by including the | Changed to "Stable oxygen and carbon isotopes of foraminiferal calcite were measured on groups of whole specimens different from those used for LA-Q-ICP-MS, using an automated carbonate |

| | | |
|---|---|---|
| | shells, the authors should specify if the isotopes were also measured on final chambers or whole shells, on single shells or multiple shells (and then how many) and whether the same shells as for Mg/Ca were used or different shells. This has all implications for the understanding of the origin of the apparent noise in the measurements. | previously missing information:
*"Stable oxygen and carbon isotopes of foraminiferal calcite were measured on groups of whole specimens different from those used for LA-Q-ICP-MS, using an automated carbonate device (Thermo Kiel IV) which was connected to Thermo Finnigan MAT 253 Dual Inlet Isotope Ratio Mass Spectrometer (IRMS)."* | device (Thermo Kiel IV) which was connected to Thermo Finnigan MAT 253 Dual Inlet Isotope Ratio Mass Spectrometer (IRMS)" |
| 2.10 | Figure 1: I agree that the two regressions (correctly using a total least squares approach) are similar, but could the authors please provide a formal statistical test for the similarity of the slopes, to support their statement that the sensitivities are indistinguishable, and for the equality of the intercepts, to dispel the impression that the regression lines are offset, indicating different endmember composition? Also, I am not convinced that it is correct to consider the results of Gat et al. (1996) as being different, as all of their values fall within the range of the presented data. | We agree that a formal statistical test for the similarity of the regressions shown in figure 1 would greatly support our argument and will therefore include the results of this test in the manuscript. We would like to thank the reviewer for this suggestion.
We understand that both datasets (Gat et al. (1996) and the data presented in our manuscript) could be considered not different from each other since ours includes "Gat-type" data points, too, we have now specified better what we meant by "different" in the original version of the manuscript. This part of the figure description of figure 1 now reads:
*"In both areas the relationship is different from the observations made by Gat et al. (1996), whose dataset suggested no statistically significant relationship between δD and δ18O of the sea water (p-value > 0.05)."* | Changed caption of Fig 1 to "The δD of the Mediterranean surface sea water is positively correlated with the local $\delta^{18}O$. The orthogonal regression of the western Mediterranean can be described as $\delta D_{water}$ =4.72*$\delta^{18}O_{water}$+1.67 (dark green). The eastern Mediterranean is very similar to the western basin, the relationship between sea water $\delta^{18}O$ and δD is $\delta D_{water}$ =5.19* $\delta^{18}O_{water}$+1.68 (light green) here. Statistically they cannot be told apart. This was determined using a bootstrapping approach that generated 100 slopes and intercepts for both the eastern and the western dataset and subsequent t-testing using the mean and standard deviation of both groups of slopes and intercepts, which resulted in p-values > 0.05. In both areas the relationship is very different from the observations made by Gat et al. (1996), whose dataset suggested no statistically significant relationship between $\delta^{18}O$ of the sea water (p-value > 0.05). " |
| 2.11 | Figure 2: Could the authors please state which regression has been used here and also provide a formal test for the lack of difference in the east and west and for the presence of a difference in the slope and intercept between their | We used ordinary least squares regressions for figure 2, assuming a linear response model. We have added the missing $R_2$-values to the figure description. The adjusted $R_2$ is 0.48 for the regression based on previously published data, | Changed caption of Fig 2 to "Surface sea water $\delta^{18}O$ is positively correlated with sea surface salinity in the Mediterranean Sea, the relationship observed can be described as linear regression $\delta^{18}O_{water}$=0.17*S-5.39 (p-value < |

| | | | |
|---|---|---|---|
| | data and literature data? Also please provide R2 for all regressions in the figure caption and/or text. | and 0.17 for the regression based on our data.
When analyzing the sampling locations (east and west) separately, the regression for the eastern samples is statistically insignificant (p-value > 0.05), likely due to the large amount of scatter and small range in salinity values compared to the overall dataset. The western part of the dataset can be described as $\delta^{18}O_{water}=0.15*S-4.75$ (p-value < 0.001, adjusted $R_2=0.06$). The results of a one-way ANOVA show that these two subsets of the data set are significantly different (p-value < 0.05), we still decided to combine them in this case, since the range of salinities is very limited for the eastern part. We have now included these information in the caption of figure 2.
We will perform a one-way ANOVA to show the difference in slope and intercept between our data and literature data, and include the result in the caption of figure 2. | 0.001, adjusted $R^2=0.17$). Previously published data can be combined into one dataset with a similar relationship with a slightly steeper slope, that is offset towards relatively higher δ18O ($\delta^{18}O_{water}=0.22*S-7.19$; p-value < 0.001, adjusted $R^2=0.48$). The two regression lines are significantly different from each other (ANOVA p-value < 0.01)." |
| 2.12 | Line 145: Considering that seawater oxygen isotopes and salinity only correlated with R2 od 0.2, the authors need an explanation for what the isotopes in foraminifera correlated more strongly with both variables. Could it be that each of the variables explains a different part of the total variance? Then, a multiple regression of foraminifera isotopes against seawater isotopes an salinity should explain significantly more variance. If it does not, it means that the two explanatory variables explain the same amount of variance. This could be because of a | We have now calculated the adjusted $R^2$ values for the regressions shown in Figure 3. They are 0.24 for $\delta^{18}O_{foraminifera}$ vs $\delta^{18}O_{water}$ and 0.42 for $\delta^{18}O_{foraminifera}$ vs salinity, these are now stated in the figure caption. We will include a paragraph discussing these in the manuscript, as well as an adjusted $R^2$ value for the relationship between sea water salinity and $\delta^{18}O$ from the subset of water samples used for the calculations shown in figure 3, following the reviewer's suggestions. | Changed caption of Fig 3 to "*G. ruber albus* $\delta^{18}O$ measurements are positively correlated (p-value < 0.001) to both sea water $\delta^{18}O$ (a) and salinity (b). The relationships can be described using the following equations: $\delta^{18}O_{foraminifera}=0.28*S-10.59$ (adjusted $R^2 = 0.42$) and $\delta^{18}O_{foraminifera} =0.95* \delta^{18}O_{water}-0.89$ (adjusted $R^2 = 0.24$). The relationship between $\delta^{18}O_{water}$ and salinity in this subset of samples is linear and comparable to that of the entire dataset ($\delta^{18}O_{water} = 0.13*S – 3.91$; p-value < 0.05, $R^2 = 0.37$). " |

| | | | |
|---|---|---|---|
| | fortuitous choice of sampling and the authors should thus also calculate the R2 for salinity and seawater isotopes only for the samples shown in Figure 3. | | |
| 2.13 | Figure 4: Could the authors again specify what regression has been used and how exactly the regression lines were calculated (regression of individual values or of the means)? Please state R2 for all regressions. Also, the Mg/Ca to T relationship is known to be exponential, so why not fitting an exponential curve? The linearity of the relationship could simply reflect the fact that the regression is fitted over a relatively narrow temperature range. | For figure 4 we used ordinary least squares regressions and included all individual data points instead of using just the mean values. The adjusted $R_2$ values of 0.13 for the salinity to Na/Ca calibration, and 0.07 for the relationship between Mg/Ca and temperature, due to the large scatter. We originally chose to use a linear regression in this case even though we are aware that an exponential curve is probably more correct, as also used for figure 6. We will fit an exponential model to this data and will update the figure caption as well as the text accordingly. | Fig 4 has been replaced, the new version shows an exponential instead of a linear fit.

Caption of Fig 4 changed to "(a) Na/Ca measured *in G. ruber albus* F-chambers collected as living specimens from the eastern and western Mediterranean Sea correlates well with local salinity (p-value < 0.001, Na/Ca=0.60 * S-13.84), even though a large natural spread of elemental composition around the mean values per station exists ($R^2$=0.13). For salinities with more than 5 individual Na/Ca measurements, hollow circles with whiskers indicate average values and standard deviations. (b) Mg/Ca in F-chambers of *G. ruber albus* specimens collected from the water column of the Mediterranean Sea is positively correlated with sea surface temperature and can be described with the exponential relationship Mg/Ca = 0.37*exp(0.14*T). For temperatures with more than 5 individual Mg/Ca measurements, hollow circles with whiskers indicate average values and standard deviations. Regression lines were calculated using all individual data points." |
| 2.14 | Line 155: Considering that Mg/Ca is also changing as a function of salinity, why not plotting Mg/Ca against salinity and analyzing the strength of that relationship as well? | Since temperature and salinity are co-varying strongly in the Mediterranean Sea, any relationship between Mg/Ca and salinity obtained from such analysis would be heavily influenced by temperature and thus appear stronger than it actually is. | No changes made. |
| 2.15 | Line 160: it is true that the foraminifera may have | We agree with all points brought up here, the | No changes made. |

| | | travelled a long distance over the 30 days of the simulation, but I question the significance of the so derived variability for the interpretation of the shell geochemistry. Culturing observations indicate that G. ruber in the size range as analysed here produces a new chamber about every two days. Thus, the particle tracking result has no bearing on the laser-ablation data. For the isotope data, if we assume a total lifespan of 4 weeks and a life expectancy of the specimens in the analysed size range of two weeks, then the collected specimens would have only had two weeks to grow, not 30 days. On top of that, because of the exponential growth of the shell, almost all of the analysed calcite and thus almost all of the isotopic signal is present in the last few chambers of the shell, so it reality, the backtracking relevant to the analysed signal should not have been carried back for more than a week. This is not to say that the result stated here is wrong – it is just that the result is not relevant for the interpretation of the measured geochemical signals. I note that your discussion in 4.4.1 resonates well with what I write, but then I do not really understand what was the merit or the justification of showing the particle backracking results in figure 7 over 30 days? | geochemical imprint of environmental parameters experienced longer ago have little impact on the bulk shell geochemistry compared to more recently experienced conditions, due to the strong increase in chamber size during foraminiferal growth. We still decided to show the trajectory for the full 30 days, which might exceed these specific specimens' lifetimes, as a worst case scenario that could be transferred to studies using larger specimens from sediment, for example. We also think that while the impact is small, it still contributes to the overall puzzle and needs to be addressed. The more factors and impacts can be quantified (and even better if they turn out to be small!), the more we become aware of the limitations of paleoclimate proxies and can make informed decisions on whether they can be applied confidently or not. | |
| 2.16 | Line 170 and onwards: please see the comments above as to the necessity to provide statistical tests to support the presence or absence of differences in regression shapes. Also, please consider | We agree that statistical tests are needed to conclusively show the difference between the data presented by us and the literature data. We will report the results of these tests in the caption of figure 2 | Changed caption of Fig 2 to "Surface sea water δ18O is positively correlated with sea surface salinity in the Mediterranean Sea, the relationship observed can be described as linear regression |

| | | |
|---|---|---|
| | the location of the sampling by Gat and yours: what if the apparent offset from your regression that he reports simply reflects the fact that he sampled at locations where the relationship is unusually confounded by secondary variables and that your data would detect the same if you only had measurements at those locations? I am also concerned by the origin of the lower oxygen isotope values measured for the given salinity in your data: was the sampling method comparable between your data and those of the previous studies (collecting from the same depth)? | and include a statement about it in the Discussion section 4.1 The majority of sampling locations used by Gat et al (1996) are nearby sampling locations used during our cruises. We have though considered that spatial differences in the sampling campaigns could contribute to the observed differences and have mentioned this in the discussion now. The updated section reads: *"Potentially the observations of Gat et al. (1996) were hence either related to unusual conditions, spatially restricted features not covered by our sampling locations or the hydrological cycle in the eastern Mediterranean has recently changed considerably"* We have carefully selected data presented in previous publications to only reflect surface waters to ensure comparability since we had also sampled at 5m water depth. Therefore different water depths do not play a role in the differences observed in the data. | $\delta^{18}O_{water}=0.17*S-5.39$ (p-value < 0.001, adjusted $R^2$=0.17). Previously published data can be combined into one dataset with a similar relationship with a slightly steeper slope, that is offset towards relatively higher $\delta^{18}O$ ($\delta^{18}O_{water}=0.22*S-7.19$; p-value < 0.001, adjusted $R^2$=0.48). The two regression lines are significantly different from each other (ANOVA p-value < 0.01)." Changed text to "Potentially the observations of Gat et al. (1996) were hence either related to unusual conditions, spatially restricted features not covered by our sampling locations or the hydrological cycle in the eastern Mediterranean has recently changed considerably" |
| 2.17 | Line 202 and onwards: Considering all your specimens were collected from the surface and that you measured only the composition of the final chamber, would it not be logically at this place to reject some of the hypotheses that you list here? Otherwise, you would have to imply that the specimens migrate vertically tens of meters over a few days, or stay alive without adding new chambers for weeks to allow lateral transport to have an effect. So perhaps we are left with the variable biomineralisation as | We agree and have changed the end of this section to address these comments. It now reads: *"Since specimens used here were collected from surface waters and add new chambers very frequently, vertical migration into water depths with significantly different conditions as suggested by Mezger et al. (2018) and Van Sebille et al. (2015) appears to be an unlikely cause for heterogeneity between specimens in this case."* | Added "Since specimens used here were collected from surface waters and add new chambers very frequently, vertical or literal migration into waters with significantly different conditions as suggested by Mezger et al. (2018) and Van Sebille et al. (2015) appears to be an unlikely cause for heterogeneity between specimens in this case." to section 4.2 |

| | | | |
|---|---|---|---|
| | the only remaining candidate mechanism? | | |
| 2.18 | Line 216: I fear the Mg/Ca data are revealing more than what the authors imply. Firstly, since the authors have both temperature and salinity, they should derive the correction independently of Gray et al. (2018) or at least check if the relationship they obtain holds. Second, I wonder why the authors do not discuss the fact that once the salinity effect is removed, their Mg/Ca data are no longer correlated with temperature or if correlated then with a much steeper slope (at least this is what I see looking at Figure 8). Third, I do not agree with the statement that the corrected values are slightly higher than expected based on the global regression – I observe that they are all higher than predicted by the exponential regression (the linear regression in Figure 8 is in my opinion superfluous). Why is that? Could there be a salinity-temperature interaction affecting the salinity-Mg/Ca relationship? This is an important result that deserves some more thought. | Since temperature and salinity are very strongly, positively correlated in the Mediterranean Sea, it is unfortunately not possible to use our data set to disentangle the effect of salinity on foraminiferal Mg/Ca from other factors such as temperature, and we therefore chose to correct for this using the equation published by Gray et al. (2018), which also increases comparability with their data set. It is indeed possible that the effect of salinity is currently underestimated, which would explain why our Mg/Ca values appear to be fairly high. We have expanded our discussion to include this consideration in our manuscript. We also have removed the word "slightly" from our description of the low temperature values. This section now reads: *"After normalizing Mg/Ca values to a sea water salinity of 35, using the calibration of Gray et al. (2018), the dependency of the Mg/Ca on temperature is similar to previously reported calibrations (e.g. Gray et al., 2018), although the Mg/Ca values at the lower most temperatures appear to be higher than expected (Fig. 6). This could potentially be caused by a combination of an underestimation of the salinity effect in these highly saline waters, since salinities observed here are well outside the calibration range used by Gray et al. (2018), and low temperatures, impacting the foraminiferal Mg/Ca comparatively little."* | Changed text to "After normalizing Mg/Ca values to a sea water salinity of 35, using the calibration of Gray et al. (2018), the dependency of the Mg/Ca on temperature is similar to previously reported calibrations (e.g. Gray et al., 2018), although the Mg/Ca values at the lower most temperatures appear to be higher than expected (Fig. 6). This could potentially be caused by a combination of an underestimation of the salinity effect in these highly saline waters, since salinities observed here are well outside the calibration range used by Gray et al. (2018), and low temperatures, impacting the foraminiferal Mg/Ca comparatively little." Fig 6 has been replaced, the new version does not include the linear regression anymore. |

| | | From context we assume the reviewer meant to refer to the linear regression in figure 6 instead of 8. We had erroneously referred to figure 6 as figure 8 in line 219 of the original manuscript, but have corrected this now. We agree that the linear regression shown in Figure 6 is not necessary, will remove it and adjust the figure caption. | |
|---|---|---|---|
| 2.19 | Line 226 (and some figure captions): please make sure species names are always written in italics | We have carefully checked the manuscript to ensure all species names are now in italics. | Changed font to italics. |
| 2.20 | Line 228: an argument on the presence (production) of G. ruber in different seasons in the Mediterranean would benefit from references to sediment trap data. There is a nice long time series from the west (Rigual-Hernandez et al., 2012) and a new dataset from the east (Avnaim-Katav et al., 2020, Deep-Sea Research) that could be used to support these statements. | We have now included the suggested references Rigual-Hernandez et al. (2012) and Avnaim-Katav et al. (2020) in this section of the manuscript and would like to thank the reviewer for bringing these studies to our attention. | Changed text to "Although low densities were reported previously for *G. ruber albus* in the Mediterranean Sea during winter time, including being absent in large areas (Pujol and Grazzini, 1995; Bárcena et al., 2004) our finding implies that lowest values in Mg/Ca can be related to winter temperatures. *G. ruber albus* is not only present throughout the year as also shown by Rigual-Hernández et al. (2012) and Avnaim-Katav et al. (2020), but it also registers the in-situ temperature, also during seasons which are close to its lower temperature limit" |
| 2.21 | Sections 4.4.1 and 4.4.2: I believe the authors could do better in providing quantitative constraints on the strength of the processes invoked to explain the large deviations in trace metals and oxygen isotopes from the theoretical calibration curves. For example, in section 4.4.2 they seem to imply that the oxygen isotope signal should be much less affected by the individual variability, but not by lateral transport. Notwithstanding of what the value of the 30-day calculation is, one should then ask: how much lateral transport would | The majority of the scatter observed in foraminiferal $\delta^{18}O$, Mg/Ca and Na/Ca likely does not stem from lateral transport, but appears to be an issue inherent to foraminiferal biomineralization. We do agree with the reviewer though that this aspect deserves more consideration and explanation and will include an extra paragraph in the discussion to cover these questions. | We added the following paragraph within 4.4.3: " It is important to note that the scatter in the foraminiferal chemistry can only to a small degree be explained by lateral transport (Fig. 7). This effect may be larger in areas where the environmental conditions vary more strongly over the distance travelled by the foraminifer, and/or in basins where there is simply more lateral transport over the foraminifer's lifetime. In our exercise, the calculated trajectories add only a minor component to the uncertainty in T (often within 0.75 °C; Fig. 7) and |

| | | | |
|---|---|---|---|
| | be needed at each of the locations to explain the isotopic scatter? Where would the calcification have to occur? Is the offset due to lateral transport large enough or not to be considered the main mechanism behind the scatter. Similarly, if all other other processes do not act on oxygen isotopes then the scatter in isotopes (residuals) should be less than in the Mg/Ca. Is it? I feel the authors should take the discussion further and provide at least first-order assessment of the strength and direction of the invoked processes and evaluate the plausibility of those processes in explaining the scatter. | | salinity (often within 0.25 salinity units)." |
| 2.22 | Line 265: on the same note: why is the lack of correlation "likely" due to all those uncertainties? How big are these uncertainties exactly? The reader needs to see the values to be able to evaluate statements like on line 271, which are intuitively correct, but not really supported by any calculations. Please provide R2 and p for both regressions shown in Figure 8. Also, the method by which the oxygen isotopes in seawater have been estimated is not sufficiently documented. For example, it is not clear if and how the salinity effect on Mg/Ca has been considered. | We have adjusted the section and replaced "likely" by "could be caused". We have also added a quantification of the quality of the two different reconstructions from Figure 8 to the main text of the discussion, by presenting the residual sum of squares (comparing reconstructed values to measured values, thus using residuals of the 1:1 relationship, not the regression lines), to support our statements made in line 271 of the original version of the manuscript. We have added the missing p-value and $R^2$ value for the relationship between measured $\delta^{18}O_{water}$ and reconstructed $\delta^{18}O_{water}$ using in situ measured temperatures. They are < 0.05 and 0.37 respectively. We agree that we did not describe the calculations well enough, we have added explanations about this to the discussion section of the | Changed text from "likely" to "could be caused". Added "The sum of squares of the residuals (difference between reconstructed and measured values) is 9.04 when using temperatures derived from Mg/Ca and $\delta^{18}O_{foraminifera}$, but only 3.56 when using temperatures measured in situ, indicating a better reconstruction." to section 4.4.3. Changed caption of Fig 8 to " The relationship between $\delta^{18}O_{seawater}$ measured in the Mediterranean Sea and $\delta^{18}O_{seawater}$ calculated from foraminiferal geochemistry (*G. ruber albus*). The relationship shown with dashed lines and cross shaped markers represents values calculated using foraminiferal $\delta^{18}O$ as well as Mg/Ca as an additional temperature proxy to decouple the effect of temperature and salinity on δ18O. The relationship shown with the continuous lines and circular markers shows the |

| | | | |
|---|---|---|---|
| | | manuscript. We did not correct for salinity in this case to avoid circular reasoning, since it is the aim of this section of the manuscript to reconstruct salinity, we therefore treated it as unknown. | same samples, but instead of using temperature values derived from foraminiferal Mg/Ca ratios, in situ measurements for temperature were used, the relationship can be described as $\delta^{18}O_{water\_reconstructed}=2.62(\pm0.69)*\delta^{18}O_{water\_measured}-63.99(\pm26.11)$ with an adjusted $R^2$ of 0.37. The temperature gradient was 2.2°C."

 Added "The Mg/Ca values used here were not corrected for salinity effects, since salinity is the target parameter that has to be reconstructed and is thus treated as unknown." |
| 2.23 | Line 281: why do the authors not take this opportunity to compare the performance of Na/Ca and the combined isotope and Mg/Ca on the resulting salinity estimates? There is no need to end with a general statement, when the authors have all the data to carry out the comparison. | We will include a section comparing the two methods, as well as describing potential issues with the use of Na/Ca as a proxy for paleosalinity. | Added "If salinity is reconstructed from the Na/Ca measurements using the calibration published by Mezger et al. (2016) and compared versus salinity measured in situ in the Mediterranean Sea, the reconstructed salinity follows the in situ measurements closely almost 1:1. The largest deviation from this 1:1 relationship occurs in the lower salinity range, at a salinity of 36.52 the reconstructed salinity estimates underestimate salinity by 0.71 salinity units. The average difference between in situ salinity measurements and salinity reconstructed based on one single-chamber measurement is an underestimation of salinity by 0.46 salinity units.
 This is still higher than the theoretical uncertainty associated when combining foraminiferal $\delta^{18}O$ and temperatures derived from Mg/Ca measured at exactly the same specimens (Rohling, 2007). An uncertainty (1SD) of 1 °C in the Mg/Ca-temperature calibration (which may be particularly optimistic at high seawater temperatures), would result in an uncertainty of ~0.37 units for the reconstructed difference between two salinities. |

| | | | This approach will lead to an improved salinity reconstruction when the (change in) past temperatures are determined more precisely, for example by reducing the error through increased sample size. The same applies for salinity reconstructions based on Na/Ca, for which not many calibrations are available and hence, leaves room for improvement. While these reconstructions as well as the lack of a strong temperature effect are very encouraging results for the use of Na/Ca as a salinity proxy, the incorporation of Na into foraminiferal calcite does not appear to be homogenous across the entire shell. It has been shown that the majority of Na in *G. ruber albus* is located in the spines (Mezger et al., 2018a, 2018b), which are not well preserved in the fossil record." to section 4.2 |
|---|---|---|---|

---

## Author Response (AR2)

Changes made to the manuscript:

A DOI has been added to point towards the data set discussed in the manuscript. The data will be made available under this DOI upon acceptance of the manuscript.

[revised manuscript text omitted]